# Association of Mastitis and Farm Management with Contamination of Antibiotics in Bulk Tank Milk in Southwest, China

**DOI:** 10.3390/ani12233392

**Published:** 2022-12-02

**Authors:** Tingrui Zhang, Sukolrat Boonyayatra, Guoyi Niu

**Affiliations:** 1College of Veterinary Medicine, Yunnan Agricultural University, Kunming 650201, China; 2Department of Food Animal Clinic, Faculty of Veterinary Medicine, Chiang Mai University, Chiang Mai 50100, Thailand; 3Faculty of Animal Science and Technology, Yunnan Agricultural University, Kunming 650201, China

**Keywords:** mastitis prevalence, bulk tank milk, farm management, antibiotic residues, risk factors

## Abstract

**Simple Summary:**

Antibiotic contamination of animal food sources is an important public health problem between animals and humans; milk is an important source of protein for people and one of the foods with antibiotic residues. Antibiotics residues in milk often come from antibiotic treatment of cow mastitis. Through a questionnaire survey study, we found that farmers on most smallholder farms have poor awareness of stewardship and disease management. Some farmers cannot accurately answer the types and usage of antibiotics. Our study also showed that there was a big gap in management between large and smallholder farms in Southwest China, which is the main reason for the prevalence of mastitis in dairy cows and the antibiotic residue rate in bulk tank milk. Our results suggest that dairy farmers need to improve their knowledge of dairy cow diseases and antibiotic use; for non-professional farmers, we recommend listening to the treatment options provided by a professional veterinarian.

**Abstract:**

Bovine mastitis could reduce the milk production and the quality of the bulk tank milk (BTM). Antibiotic treatments through intramammary or parenteral methods are being widely used in dairy farms. A cross-sectional study to investigate for general farm management and pre-test the questionnaire was performed in Southwestern Yunnan province, China. A total of 134 dairy farms were included. Milking cows of each farm were determined for the presence of clinical (CM) and sub-clinical (SCM) mastitis using the California Mastitis Test (CMT). Rates of CM and SCM in studied farms ranged from 2–11%, and 24–69%, respectively. The incidence of antibiotic residues in BTM of all farms was very high (32%, 44/134). All antibiotic contaminated samples were from smallholder dairy farms. Factors significantly associated with the presence of antibiotic contamination included farm region, antibiotics usage, persons performing mastitis treatment, and rates of CM. Rates of CM were significantly associated with the farm region, cleanliness of udders before milking, and the number of milking cows. Our results emphasize that the risk factors of dairy farm management should be paid attention, which can reduce mastitis prevalence and antibiotic contamination in BTM in Southwestern China.

## 1. Introduction

Bovine mastitis is defined as an inflammation of the mammary glands of dairy cows [1,2]. It can be infectious, caused by intramammary infection of microorganisms, or non-infectious, as a result of physical injury to the mammary gland [1,2]. The inflammatory response results in an increase in blood proteins and white blood cells in the mammary tissue, which can reduce the quality of dairy products [3,4]. A study showed that the mean cumulative incidence of CM was 3.3 per 100 cows per month in large-scare dairy farms in China [5]. However, there are few studies on mastitis that is caused or exacerbated by smallholder dairy farmers in China. Thus, investigating the prevalence of CM with different farm systems is needed to allow the implementation different therapeutic and preventive management strategies.

Bovine mastitis can lead to a decrease in milk production and early culling of dairy cows [6]. In the United States, economic losses associated with bovine mastitis have increased from USD 110 per cow per year in 2002 to USD 325 per cow per year in 2008 [7]. In China, one study reported the economic losses with clinical mastitis between USD 12,000 and 76,000 per farm per month, and between USD 29 to 135 per cow per year [8]. Additionally, the nutrient and chemical properties along with microbiological hygienic qualities of the affected milk can ultimately be unsatisfactory [9]. Management factors can influence the occurrence on mastitis on farms [10]. Therefore, understanding the prevalence and management in dairy cattle can help to create a strategic plan and implementation of preventive and control measures [11].

Antibiotic residues in milk usually come from the antibiotics used for the treatment of lactating cows [12,13]. Imprudent use of antibiotics in livestock is very much a concern in China [14,15,16]. Milk contaminated with antibiotics may, directly and indirectly, affect human health. Antibiotic residues in milk can directly trigger an allergic reaction in some people [17]. Moreover, the long-term consumption of milk containing antibiotics is undoubtedly equal to long-term use of small doses of antibiotics, which can lead to bacteria resistant to therapeutic antibiotics [18]. Antibiotic resistant bacteria are known to be associated with low cure rates of antibiotic treatment in patients, causing a high rate of chronic infections [19]. Studies in China suggest that milk is the most important source of antibiotic residues in the human body and is related to how often people consume milk [20,21]. In dairy processing, antibiotic residues can seriously influence starter culture for cheese, yogurt, and other dairy products, leading to heavy economic losses [22].

Therefore, controlling antibiotic contamination in milk should be prioritized. Since antibiotics in dairy farms are mostly used for treating bovine mastitis, we should begin by describing the association of high prevalence of bovine mastitis and the presence of antibiotic residues in milk. The levels of antibiotic residues in milk should be reduced when farmers recognize the importance of controlling antibiotic use in their farms through improving management to lower mastitis problems in their farms [23]. The ultimate goal is to improve dairy food safety for the consumer.

In order to better control and monitor veterinary drug residues in food, the Ministry of Agriculture and Rural Affairs of China (MARA) has issued maximum residue limits (MRLs) for veterinary drugs in food in 2002 and made a revision in 2019. MARA issued regulations for the administration of antibiotics in 2004 and last amended them in 2018. The regulated MRLs for common antibiotics used in veterinary medicine in China include some common veterinary antibiotics in milk, such as β-Lactam, Sulfonamide, Tetracycline, Macrolide, and Aminoglycoside [24]. China established a risk assessment system for food safety, the China National Center for Food Safety in 2009, to better support food safety monitoring and supervision [25].

Southwest China is an underdeveloped area in China, and the per capita income is much lower than that of Eastern and Central China [26]. There is limited research on risk factors for antibiotic residues in milk in dairy farms in Southwest China. Yunnan Province is the most important and representative province in Southwest China for the livestock industry. Risk management research on dairy farms in this region will help improve public health in developing areas. The purpose of this study was to investigate the prevalence of clinical and subclinical mastitis in dairy farms in Yunnan Province and to explore the relationship between antibiotic residues in BTM and farm management.

## 2. Materials and Methods

### 2.1. Sample Collection

A three-month cross-sectional study was operated in Yunnan, China (25°3′2.00″ N, 102°42′40.20″ E, Figure 1). In this area, there are a total of 17,703 milking cows. Most dairy breeds are Holstein. There are two major types of dairy farms in Yunnan. A class includes dairy processing enterprise-owned farms. They have their own parlors; these farms supply milk for their company, respectively. Another type includes multiple owners in each community shared farm. In dairy farms in each community, different owners can share farm dairy facilities (such as milking parlors) and production of different dairy milk processing enterprises. Dairy communities consisted of some smallholder dairy farms; the standard farm is a large-scale dairy farm. This study selected 6 regions willing to participate in the survey based on random selection and compliance with the farmer’s wishes, 3 standard farms (large-scale farm), and 3 dairy communities (131 smallholder farms) were included in this study. Selected farms’ information is shown in Table 1.

The list of 6 dairy farms/dairy communities was used as a sampling frame, and the sample size from the sampling unit (number of dairy cows in each farm) was generated through online EpiTools^®^ epidemiological software developed by the AusVet Animal Health Services. A total of 745 dairy cows, out of 4,351 dairy cattle from 45 dairy farms, were selected through random sampling. The sample size of 745 is calculated given: assumed prevalence = 0.5, assumed sensitivity = 0.9, assumed specificity = 0.8, population size = 17,703, confidence level = 0.95, desired precision = 0.05

A stratified random sampling of studied farms was performed based on management system (large-scale farms vs. community farms), and farm size (large vs. small) using median number of milking cows, to sample 6 farms consisting of 3 company and 3 community farms. For the CMT criteria, in community farms, when a farmer has <10 cows test all, if a farmer has 10–50 cows test 50%, if a farmer has 50–100 cows test 20%, if a farmer has >100 cows test 10%. For a company, test 10%. A total of 745 samples were obtained.

### 2.2. Farm Information Collect

A questionnaire was designed for an interviewer survey to collect farm data. General farm data were collected by face-to-face interviews. These data included farmer’s age, farmer’s name, farmer’s nationality, farmer’s education level and experience, knowledge of the farmer about subclinical mastitis, drugs used in farm, sick animal management, drug knowledge of farmer/worker, and antibiotic testing before selling milk. Management data associated with milking hygiene were collected by observation. These observed data included cleanliness of barn and pipeline milking system, and cleanliness of udders before milking. Udder cleanliness was scored based on the scoring chart described by Munoz [27]. All data were collected by only one professional veterinary interviewer/observer.

### 2.3. Clinical and Subclinical Mastitis

Each farm was determined for the numbers of clinical mastitis (CM) cows once per month in order to get an averaged number of CM cows per month. During visits, cows showing clinical signs of mastitis including udder swelling, udder firmness and warmness, and abnormal milk appearance with/without other systemic signs were observed and defined as clinical mastitis cows.

All milking cows without any clinical signs on each farm were tested using the California mastitis test (CMT) during milking time to identify subclinical mastitis (SCM) cows. CMT was performed on each selected cow by mixing approximately 2 mL of quarter milk to the same volume of CMT reagent. Quarter milk with observed reaction of a small amount of viscous precipitate at the bottom of the testing plate was considered to be from a mastitis udder. Cows having at least 1 udder producing mastitis milk were defined as cows with subclinical mastitis.

### 2.4. Antibiotic Residues in Milk

Bulk tank milk (BTM) from each farm was collected and tested monthly for a total of 3 times per farm to detect antibiotic residues. For smallholder dairy farms, composite milk samples from all milking cows were mixed to represent a BTM sample for each farm. For large-scale dairy farms, BTM was sampled directly from their cooling tanks. Fifty milliliters of BTM were collected at each visit. BTM samples were stored on ice and transported to the laboratory within 24 h.

Antibiotic residues in BTM were detected using Delvotest^®^ SP-NT (DSM, Heerlen, Netherlands). Regarding the manual, 0.1 mL of BTM was added into the tube containing spores of *Bacillus stearothermophilus*, agar medium and bromocresol purple as a pH indicator. The tubes were incubated at 64 ± 2 °C for 3 h. Milk samples were considered positive when color change of the media was not observed. Tubes with color-changed media from purple to yellow were considered negative.

### 2.5. Data Analysis

Prevalence of CM and SCM at the animal level was calculated (%). Detection rates (%) of antibiotic residues in BTM were calculated for each sampling. Incidence (%) of antibiotic residues in BTM for the 3-month period was calculated as follows: (number of farms having at least 1 Delvotest-positive BTM in 3 months/number of total farms) × 100. Data analysis is described by a popular method in different sampling units, the odds ratio was used to find factor variables associated with statistically significant results. Statistical analysis version 3.0.3 R statistical software program were used to describe the relationship between dairy farm management and milk antibiotic risk factors. Logistic regression and multiple linear regression analyses were conducted using backward stepwise analysis in R statistical program [28]. Factors added in the initial model are listed in Table 2. Farm geographic information was analyzed by QGIS 2.18.

## 3. Result

### 3.1. The Questionnaire Surveys

In Yunnan province, the majority of dairy farms were smallholder dairy farms, which were usually grouped together in several dairy communities. A dairy community was composed of a number of smallholder dairy farms, and farmers are both the owners of the farms and farm workers. Most farmers were between 40 to 50 years old. Dairy farmers in this study had relatively low educational level or were uneducated. University graduates accounted for only 2% and worked in the standard farms (Table 3). About half of the studied farmers (52%) had not received farm training.

Most dairy farms (131 farms) purchased their cows from local providers. One farm imported their cows from New Zealand, and another farm imported their cows from Canada (Table 3). All of the studied smallholder dairy farms’ management systems were tied stalls with concrete floors containing different bedding materials, whereas all large-scale dairy farms’ management systems were free stalls (Table 3).

Concerning the management of sick animals within the farms, most farmers did not have a system to identify sick and treated animals within the farms (66.41%) (Table 3). In addition, all studied farms had penicillin-streptomycin for treating animals in their farms. Some other antibiotics were also observed in some farms (Figure 2).

### 3.2. The Prevalence of Mastitis

Smallholder dairy farms in dairy communities E and F had the highest prevalence of CM (11%). Lower prevalence of clinical mastitis was observed among large scale dairy farms (2% to 6%) compared to smallholder dairy farms (6% to 11%) (Table 4). Regarding the prevalence of SCM in the studied farms, smallholder dairy farms in community D showed the highest prevalence (69%), followed by farms in community E (53%) and community F (52%) (Table 4).

### 3.3. Antibiotic Residues in Milk

Detection rates of antibiotic residues in BTM for the 1st, 2nd, and 3rd samplings were 8.95%, 13.43% and 16.42%, respectively (Table 5). All antibiotic contaminated samples were from smallholder dairy farms. Regarding the overall 3 sampling times in the 3-month period, the incidence of antibiotic contamination in BTM was 32% (Table 5).

### 3.4. Factors Associated with the Presence of Antibiotic in BTM

After logistic regression analysis, 4 variables remained in the final logistic regression model: farm regions A-F, only use penicillin-streptomycin, the person performing mastitis treatment, and rates of clinical mastitis as shown in Table 6. After analysis, 3 variables remained in the final multiple linear regression model: farm regions A-F, cleanliness of udders before milking, and the number of milking cows as shown in Table 7.

## 4. Discussion

### 4.1. General Information about the Farms

Some core attributes, for example, farmer productivity, management of sick animals, and treatment, are particularly weak aspects of the dairy production system of smallholder dairy farms in Yunnan. These three attributes not only influence the other attributes but also have long term implications for sustainability. Poor quality of animal management adversely affects milk productivity and hence the cost of milk production and productivity of other inputs. Thus, the formulation and planning of farm management strategies to improve the management of sick animals and animal treatment should be prioritized for the development of the dairy industry in Yunnan. Furthermore, to improve the farmer’s knowledge along scientific lines, development is required in the provisioning of adequate and quality livestock support services [29,30].

Farmworkers are essential for successful dairy farming. Without a reduction in the drudgery of their work, enhanced control over their financial resources, and their access to information regarding dairy farming, smallholder dairy farms will not be socially sustainable [31]. There is a need to develop operational technologies supporting smallholder farms to improve the efficiency of the workforce and reduce drudgery for the workers [32].

Large-scale dairy farms are very different from smallholder dairy farms in terms of their management, their clear vision of farming objectives, and their readiness to adopt new innovations to improve their farm management. This may be because large-scale dairy farms usually have at least one well-educated worker to lead and to design and control the productivity of the farm. Moreover, in large-scale dairy farms, the application of digital controls in monitoring the production of the entire farm and, potentially, a specific herd (mainly including reproduction and milking performance) offers the basic data for dairy herd improvement (DHI) [31,33]. This innovation not only helps to monitor the quantity and quality of raw milk, but also provides data to predict nutrient requirements for individual cattle, and serves both precision diet formulation and fine feeding according to cattle individuals [31].

### 4.2. Prevalence of Mastitis in Dairy Cows

The prevalence of clinical and subclinical mastitis in dairy farms in this study was quite high and was similar to that in several previous studies in China. For example, the prevalence of SCM in West China was 54.3% and the udder quarter basis was 54.3% [34]. In Southeast China, a study shows that the prevalence of SCM was 38.4% [35]. In Central China, the prevalence of SCM in heifers was 18.78% and the prevalence of CM was 3.86% [36]. According to reports, the global mastitis prevalence rate is 20% to 80% [37,38]. A study reported in the UK that the average annual incidence of CM during a three-year period was 43.4% [39]. In New Zealand, mastitis occurs in 14.8% of lactating cows [40].

A higher prevalence of clinical and subclinical mastitis was observed in smallholder dairy farms compared to large-scale dairy farms. Based on our data, most smallholder dairy farmers did not have a system in place to identify sick and treated animals within the farms. Many sick cows and healthy cows can make contact with each other, and may consequently affect the spread of mastitis pathogens among cows in the farms [41]. Moreover, we found that all smallholder dairy farms kept their cows on a concrete floor, which may encourage the cows to lie down for a long time, making their udders prone to infection [1,42]. According to milking hygiene, most smallholder dairy farms did not use disinfectant for cleaning cows’ udders before milking. Poor cleanliness of cow’s udders was also observed before milking. This mismanagement may affect the increased prevalence of mastitis in these farms because inappropriate udder preparation before milking, especially with respect to wet and dirty udders, can easily lead to the spread and multiplication of bacteria on the udders and teats, causing bovine mastitis [33,43].

### 4.3. Antibiotic Residues in the BTM

Many studies had surveyed antibiotic residues in milk in different regions of the world using different methods, such as Delvotest^®^ SP, Triphenyltetrazolium chloride (TTC) method and Copan test. One study detected 40.8% antibiotic residue in pasteurized milk from Iran [44]. A Serbian study found the antibiotic residue of bulk tank milk was 40% [45]. A study found that antibiotic residues in commercial and farm milk collected from Algeria were as high as 65.5% [46]. Another study from China showed that the detection rates for quinolones and sulfonamides in raw milk in 10 provinces of China were 47.2% and 20.1%, respectively [47]. We reported here that the high detection rates for antibiotic residues in selected farms in Yunnan range from 8.95% to 16.41%.

Antimicrobial-resistant bacteria are proliferating around the world [48]. China is a country with a heavy use of antibiotics in animal husbandry and fisheries [48]. This is associated with high rates of antimicrobial-resistant bacteria reported in this country. More than 60% of *Staphylococcus aureus* isolated from Chinese patients in surveyed hospitals in 2009 were methicillin-resistant (MRSA) which was an increase from 40% in 2000 [48]. In addition to MRSA, *Streptococcus pneumonia* isolated from Chinese patients also showed a high rate, up to 70%, of resistance to macrolides [48]. Moreover, approximately 60 to 70% of *Escherichia coli* are resistant to quinolones, which is the highest rate in the world [48]. These examples indicate that the health issue of antimicrobial-resistant bacterial, which can be caused by the long-term consumption of antibiotic-contaminated food, is a major concern in China.

### 4.4. Factors Affecting Antibiotic Contamination in BTM

The current study hypothesized that farms with high rates of clinical mastitis can contribute to the high use of antibiotic treatment for mastitis, which consequently is associated with increased antibiotic contamination in BTM. From the analysis of our data set, this hypothesis was confirmed. We found that farms with high rates of clinical mastitis are prone to having antibiotics contaminating their BTM.

Different farm management systems seemed to influence the rates of antibiotic contamination in BTM. In the present study, using Delvotest^®^, none of the BTM of large-scale dairy farms was found to be contaminated with antibiotics. Only the BTM of smallholder dairy farms was found to be contaminated with antibiotics. We could not provide any evidence to explain this finding in the present study. However, the low rate of antibiotic residues in milk from large-scale dairy farms may be related to the low levels of bovine mastitis on dairy farms, which indicates a low chance of using antibiotics for treatment as previously described and found in this study [49,50,51]. Therefore, controlling clinical mastitis in dairy farms by improving milking hygiene, for example paying attention to the cleanliness of the udders before milking as suggested in this study, may help to reduce the use of antibiotics in the farm, as well as decreasing the risk of antibiotic contamination in BTM.

Other management factors related to antibiotic use on farms were also found to be associated with the presence of antibiotics in BTM in Yunnan. From both observation and interviewing, it was apparent that some farms had only penicillin/streptomycin in their farms, while some farms had penicillin/streptomycin and other antibiotics. The presence of antibiotic products indicates the use of these products on the farm. Farms using different kinds of antibiotics seemed to have a greater likelihood of antibiotic contamination in their BTM, as previously described [52]. Moreover, BTM of farms whose treatments were performed by veterinarians were less likely to be contaminated with antibiotics. In 1997, Sischo et al. reported that risk factors for antibiotic residues in milk included a lack of proper treatment records, a poor understanding of how to use antibiotics, and poor relationships between farmers and veterinarians [53]. These topics should be included in the educational program for dairy farmers in Yunnan, China, to reduce the risks of antibiotic contamination in BTM. On the positive side, in 2017, China developed a national veterinarian system for training, managing, and supervising animal health workers [54], which will help to reduce antibiotic residues in milk.

## 5. Conclusions

In conclusion, the prevalence of both clinical and subclinical mastitis in the studied dairy farms in Yunnan was quite high. Similarly, rates of antibiotic residues in BTM of the selected farms in Yunnan were high. We found a big difference in the management level of dairy farming between the two types of farms: large-scale and smallholder dairy farms, in Yunnan, China, which is one of the significant factors related to both antibiotic residues in BTM and clinical mastitis. Our data set also indicates that farms with higher rates of clinical mastitis and which have farmers performing mastitis treatment are likely to have antibiotic residues in BTM. Therefore, prudent use of antibiotics in dairy farms, under the supervision of veterinarians, should be promoted in Yunnan, China.

## Figures and Tables

**Figure 1 animals-12-03392-f001:**
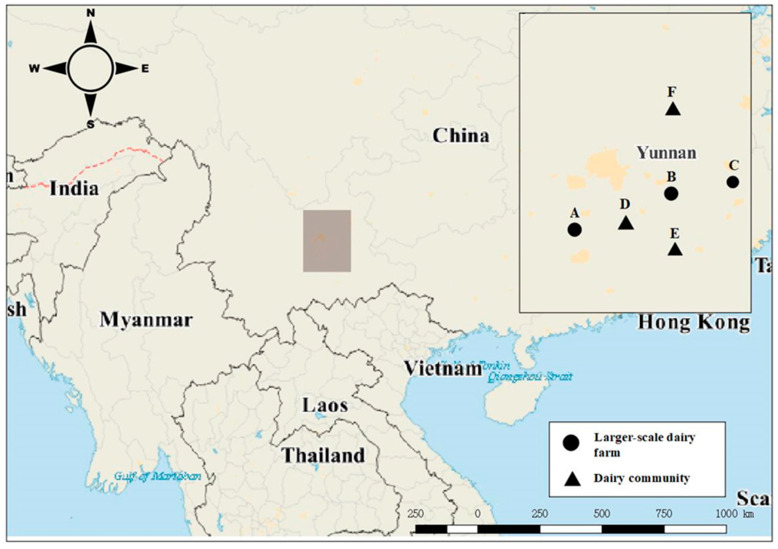
The locations of the farms collected in Yunnan, China. A: Large farm A, B: Large farm B, C: Large farm C, D: Dairy community D, E: Dairy community E, F: Dairy community F.

**Figure 2 animals-12-03392-f002:**
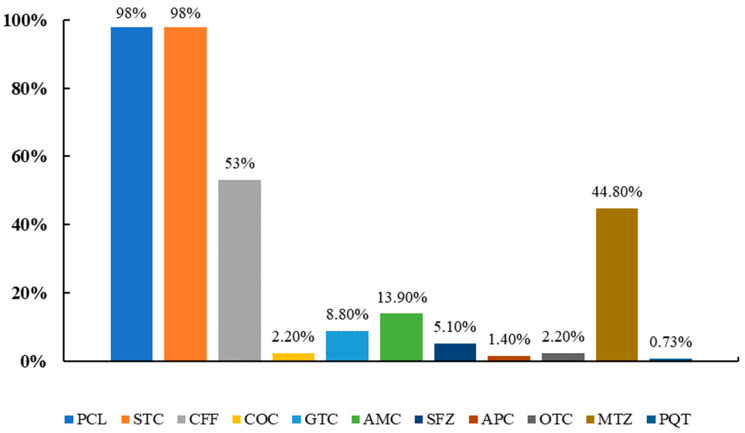
Drug usage rate in dairy farms in Yunnan, China; PCL = Penicillin, STC = Streptomycin, CFF = Ceftiofur, COC = Chloromycetin, GTC = Gentamicin, AMC = Avermectin, SFZ = Sulfadiazine, APC = Ampicillin, OTC = Oxy tetracycline, MTZ = Metronidazole, PQT = Praziquantel.

**Table 1 animals-12-03392-t001:** Basic information of selected farms to be investigated in the study.

Farm Types	Total Cows	Milking Cows	Milk Production (Ton/Year)	District
Large-scale dairy farms				
Large farm A	320	300	6.5	Jinning
Large farm B	301	101	4.5	Yiliang
Large farm C	2000	1700	7.2	Shilin
Smallholder dairy farms				
Community D (30 farms)	1040	450	4.2	Jingning
Community E (51 farms)	2600	1700	3.95	Jingning
Community F (50 farms)	2160	1000	4.8	Songming

**Table 2 animals-12-03392-t002:** Variables included in the initial logistic regression model for antibiotic residues in BTM and the initial multiple linear regression model for rates of clinical mastitis.

Variable Codes	Definitions	Logistic Regression	Multiple Linear Regression	Type of Variable
bhr	Cleanliness of udders before milking		√	Categorical
RCM	Rates of clinical mastitis	√		Continuous
CMTR	Rates of subclinical mastitis as determined by CMT	√	√	Continuous
dkof	Farmer’s knowledge regarding drug use	√	√	Categorical
el	Educational level of farmer	√	√	Categorical
floor	Floor type of stable	√	√	Categorical
hlwp	Withdrawal period usually performed in the farm	√		Categorical
ht	Person performing mastitis treatment in the farm	√		Categorical
kcmt	Farmer’s knowledge regarding CMT	√	√	Categorical
N.O.M.C.	Number of milking cows	√	√	Continuous
N.O.V.	Number of veterinarians giving service in the farm		√	Continuous
Pen-strep	Farmer having or using only penicillin/streptomycin	√		Categorical
Post-dip	Post-milking teat dipping		√	Categorical
region	Large-scale dairy farm A, B, C and smallholder dairy farms in community D, E, F		√	Categorical
sam	Sick animal management	√	√	Categorical
type	Type of farm (large-scale vs. smallholder dairy farms)		√	Binary
wrdt	Whether or not the farmer received training in dairy farming	√	√	Binary

**Table 3 animals-12-03392-t003:** Descriptive data of farmers and dairy farms included in the study.

Factors	Levels	Total Number (%)	Number of Smallholder (%)	Number of Large-Scale (%)
Ages	21–30	6% (8/134)	6% (8/131)	0% (0/3)
	31–40	28% (37/134)	27% (36/131)	33% (1/3)
	41–50	56% (75/134)	55% (73/131)	66% (2/3)
	>50	10% (13/134)	9% (13/131)	0% (0/3)
Education	No	54% (73/134)	55% (73/131)	0% (0/3)
	Primary school	22% (29/134)	22% (29/131)	0% (0/3)
	High school	22% (29/134)	22% (29/131)	0% (0/3)
	Undergraduate	2% (3/131)	0% (0/131)	100% (3/3)
Sources of cows	Local	98% (131/134)	100% (131/131)	0% (0/3)
	New Zealand	0.7% (1/134)	0% (0/131)	50% (1/2)
	Canada	0.7% (1/134)	0% (0/131)	50% (1/2)
Management system	Free stall	2.2% (3/134)	0% (0/131)	100% (131/131)
	Tied stall	97% (131/134)	100% (131/131)	0% (3/3)
Floor	Concrete floor	131/134 (97.76%)	131/131 (100%)	0/3 (0%)
	Composite floor	3/134 (2.24%)	0/131 (0%)	3/3 (100%)
Disinfectants	Use	3/134 (2.24%)	0/131 (0%)	3/3 (100%)
	Not use	131/134 (97.76%)	131/131 (100%)	0/3 (0%)
Cleanliness of udders before milking	Very dirty	17/134 (12.69%)	17/131 (12.98%)	0/3 (0%)
	Dirty	55/134 (41.79%)	55/131 (42.75%)	0/3 (0%)
	Clean	57/134 (42.54%)	54/131 (41.98%)	2/3 (66.67%)
	Very clean	4/134 (2.99%)	3/131 (2.29%)	1/3 (33.33%)
Pre-milking teat dipping	Yes	3/134 (2.24%)	0/131 (0%)	3/3 (100%)
	No	131/134 (97.76%)	131/131 (100%)	0/3 (0%)
Post-milking teat dipping	Yes	3/134 (2.24%)	0/131 (0%)	3/3 (100%)
	No	131/134 (97.76%)	131/131 (100%)	0/3 (0%)
Farmer know CMT	Know	20/134 (15.67%)	17/131 (13.74%)	3/3 (100%)
	Don’t know	114/134 (84.33%)	114/131 (86.26%)	0/3 (0%)
Sick animal management	Nothing	87/134 (64.93%)	87/134 (66.41%)	0/3 (0%)
	Mark but not separate	19/134 (14.18%)	19/131 (14.5%)	0/3 (0%)
	Mark and separate	3 (2.24%)	0/131 (0%)	3/3 (100%)
	Separate but not mark	25/134 (18.66%)	25/131 (19.08%)	0/3 (0%)
Person performing mastitis treatment	Farmer	97/134 (72.39%)	97/131 (74.05%)	0/0 (0%)
	Veterinarian	37/134 (27.61%)	34/131 (25.95%)	3/3 (100%)

**Table 4 animals-12-03392-t004:** The prevalence of mastitis in studied farms in Yunnan, China.

Dairy Farms	Prevalence of Clinical Mastitis	Prevalence of Subclinical Mastitis
Large-scale farm A	6%	24%
Large-scale farm B	2%	30%
Large-scale farm C	3%	39%
Small farms in community D	6%	69%
Small farms in community E	11%	53%
Small farms in community F	11%	52%

**Table 5 animals-12-03392-t005:** Detection rates and incidence of antibiotic residues in bulk tank milk (BTM) of selected large scale and small holder dairy farms in 3 communities in Yunnan, China.

Dairy Farms	Detection Rates of Antibiotic Residues in BTM (%)			Incidence of Antibiotic Residues in 3 Months (%)
	1st Sampling	2nd Sampling	3rd Sampling	
Large farms	0% (0/3)	0% (0/3)	0% (0/3)	0 (0/9)
Small farms in community D	4% (2/51)	9.80% (5/51)	11.54% (6/51)	21% (11/51)
Small farms in community E	14.81% (4/27)	33.33% (9/27)	29.60% (8/27)	59% (16/27)
Small farms in community F	9.43% (5/53)	7.54% (4/53)	15.09% (8/53)	32% (17/53)
Total	8.2% (11/134)	13.43% (18/134)	16.41% (22/134)	32% (44/134)

**Table 6 animals-12-03392-t006:** Variables in the final logistic regression model for the presence of antibiotic in bulk tank milk in the 3 month period 9.

Factors	Estimate	Std. Error	Odds Ratio (OR)	95% Confidence Interval of OR	*p*-Value
**Region ^1^**					
A	(Reference level)				
B	2.38	0.609	10.87	3.29, 35.92	<0.01
C	1.97	0.712	1.72	0.71, 4.15	<0.01
D	−14.42	2399.545	0	0, infinity	>0.05
E	−14.42	2399.545	0	0, infinity	>0.05
F	−14.42	2399.545	0	0, infinity	>0.05
**Pen-strep**					
Only penicillin/ streptomycin was observed or used in the farm	(Reference level)				
Other antibiotics were presented or used together with penicillin/streptomycin	1.52	0.6166	4.56	1.36, 15.28	<0.05
**Rate of clinic mastitis**	1.9472	1.4532			>0.05
**Person performing mastitis treatment**					
Farmer	(Reference level)				
Veterinarian	−0.9784	0.5231	0.38	0.13, 1.05	<0.05

^1^ farm region; large-scale dairy farm A, B, C and smallholder dairy farm D, E and F.

**Table 7 animals-12-03392-t007:** Variables in the final multiple linear regression model for the prevalence of clinical mastitis in Kunming dairy farms.

Factors	Estimate	Std. Error	95% Confidence Interval	*p*-Value
**Region ^1^**				
A	(reference level)			
B	−0.018	0.041	−0.099, 0.062	>0.05
C	−0.026	0.037	−0.099, 0.047	>0.05
D	0.364	0.197	−0.023, 0.750	>0.05
E	0.9601	0.383	0.210, 1.711	<0.05
F	5.083	1.701	1.749, 8.417	<0.01
**Cleanliness of udders before milking ^2^**				
very dirty	(reference level)			
dirty	0.057	0.038	−0.018, 0.131	>0.05
clean	0.116	0.050	0.018, 0.214	<0.05
very clean	0.039	0.090	−0.138, 0.217	>0.05
**Number of milking cow**	−0.006	0.002	−0.011, −0.002	<0.01

^1^ farm region; large-scale dairy farm A, B, C and smallholder dairy farm D, E and F. ^2^ Clean the nipple use 40 °C water and medicinal bath before milking.

## Data Availability

The authors confirm that all relevant data are within the paper.

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
