# Peer review of "Association of Mastitis and Farm Management with Contamination of Antibiotics in Bulk Tank Milk in Southwest, China"

_animals, 2022, doi:10.3390/ani12233392_

Round 1

Reviewer 1 Report

In this manuscript, the authors study the impact of different factors (presence of mastitis, antibiotic usage, cleanliness of the udder, etc) on the presence of antibiotic residue in bulk tank milk in 2 kinds of farms (large industry driven farms and community farms that regroup small farms) in Southwestern China.

It is interesting to learn how dairy industry is organized in that region.

The results obtained are exactly as expected.

The manuscript must be reviewed by a professional to improve the quality of the language.

There is a confusion between udder and quarter (L38, L138). A cow has one udder that is composed of 4 quarters.

Please provide more information about the farms especially the small ones. For example, in Figure 1, the characteristics of the 6 farms that were studied could be added. Please also add the total number of small farms in each community.

L48: This number is for Spain. Is that data available for China ? What is an average herd for Spain?

Table 2: What does the 1 in the first criterion stand for?

Figure 2: The 2 last antibiotics in the figure (MTZ and PQT) are not listed in the legends

Table 7: Rate of clinical mastitis

Author Response

Dear Reviewer,

Thank you for your reviewers’ comments concerning our manuscript entitled “Association of mastitis and farm management to the contamination of antibiotics in bulk tank milk in Southwestern area, China” (ID: animals-1949262). Those comments are all valuable and very helpful for revising and improving our paper, as well as the important guiding significance to our research. We have studied the comments carefully and have made corrections which we hope meet with approval. Our modified responses are shown below, please check them.

Response to reviewer 1

Line 38,line138. There is a confusion between udder and quarter (L38, L138). A cow has one udder that is composed of 4 quarters.

Response: From the structure of the dairy cow's udder, one udder is divided into four milk regions (one region = quarter), quarter milk region represents an independent secretion system. In our study, we defined a cow that had clinical/subclinical mastitis in a quarter region as a case of mastitis.

Please provide more information about the farms especially the small ones. For example, in Figure 1, the characteristics of the 6 farms that were studied could be added. Please also add the total number of small farms in each community.

Response: We have changed the large farm code name in Table 1, and added the total number of farms in the dairy community in Table 1. Added more information in Figure.

Line L48: This number is for Spain. Is that data available for China? What is an average herd for Spain?

Response: We remove the reference from Spain. We have added more appropriate references, which have data in China. new reference:

He W, Ma S, Lei L, He J, Li X, Tao J, Wang X, Song S, Wang Y, Wang Y, Shen J, Cai C, Wu C. Prevalence, etiology, and economic impact of clinical mastitis on large dairy farms in China. Vet. Microbiol, 2020. 242: p. 108570.

Table 2: What does the 1 in the first criterion stand for?

Response: 1 is a write error. And We deleted table 2, table 2 shows the reference standards for our other study. We have changed it to the correct reference: Munoz MA, Bennett GJ, Ahlström C, Griffiths HM, Schukken YH, Zadoks RN. Cleanliness scores as indicator of Klebsiella exposure in dairy cows. J Dairy Sci, 2008. 91(10): p. 3908-16.

Figure 2: The 2 last antibiotics in the figure (MTZ and PQT) are not listed in the legends

Response: The MTZ and PQT were added to the legend.

Table 7: Rate of clinical mastitis

Response: Change Rats to Rate

Reviewer 2 Report

Dear Authors,

Manuscript entitled “Association of mastitis and farm management to the contamination of antibiotics in bulk tank milk in Southwestern area, China” highlighted the importance of rationale usage of antibiotics in animals.

The manuscript draws attention to the issue of antibiotic overuse and suggests that more attention to controlling antibiotic contamination in milk should be prioritized. The global usage of antibiotics in animals is double compared to humans, so significant portions of antibiotics can be released through the milk of dairy animals unaltered and cause serious harmful effects on human health. Excessive antibiotic treatment of cows with mastitis has resulted in greater resistance to mastitis-associated pathogens. Mastitis is a multifactorial disease in dairy cows requiring proper herd management to eliminate or minimize its incidence and economic losses. Parts of the manuscript (Introduction, Material and Methods, and Results) are fairly well organized and carried out. The discussion encompasses the following parts (General information about the farms, Prevalence of mastitis in dairy cows, and Antibiotic residues in the BTM) are written systematically, clearly, and logically which leads to the correctly performed conclusion that is in line with obtained results. Through manuscript, science and the presentation are strong. So, I would suggest that it be accepted for publication in its present form.

Author Response

Dear Reviewer,

Thank you for your reviewers’ comments concerning our manuscript entitled “Association of mastitis and farm management to the contamination of antibiotics in bulk tank milk in Southwestern area, China” (ID: animals-1949262). Those comments are all valuable and very helpful for revising and improving our paper, as well as the important guiding significance to our research. We hope to meet with approval. 

Reviewer 3 Report

In this paper, the Authors investigate the prevalence of clinical and subclinical mastitis in dairy farms in Yunnan Province, the relationship between antibiotic residues in bulk tank milk and farm management.

The topic is very interesting and fit well within the scope of the journal.

Every day and each year lactating dairy cows receive antibiotic therapy for clinical mastitis, and to ensure food safety for consumers, several regulatory authorities around the world (i.e. EFSA, FDA,

and Codex Alimentarius) established tolerance levels of antibiotic residues (so called Maximum

Residual Limit, MRL) in milk and other foodstuffs for consumer protection.

This aspect is not considered in the manuscript. In China are there legal guidelines regarding antibiotic residues? Are MRL values are established? And who is the checker?

Please improve the introduction adding a paragraph about this legislative issue.

Furthermore, the conclusions were also presented in a reductive manner, please improve it.

A  minor spell check of  English language is required

For these reasons, the manuscript is not recommended for publication. Reconsider after major revision

Author Response

Dear Reviewer,

Thank you for your reviewers’ comments concerning our manuscript entitled “Association of mastitis and farm management to the contamination of antibiotics in bulk tank milk in Southwestern area, China” (ID: animals-1949262). Those comments are all valuable and very helpful for revising and improving our paper, as well as the important guiding significance to our research. We have studied the comments carefully and have made corrections which we hope meet with approval. Our modified responses are shown below, please check them.

Response to reviewer 3

In China are there legal guidelines regarding antibiotic residues? Are MRL values are established? And who is the checker?

Response: Added content “In order to better control and monitor veterinary drug residues in food, the Ministry of Agriculture and Rural Affairs of China (MARA) have issued the maximum residue limits (MRLs) for veterinary drugs in food in 2002, and made a revision in 2019. MARA issued regulations on the administration of antibiotics in 2004 and last amended them in 2018. The regulated MRLs for common antibiotics used in veterinary medicine in China, the content includes MRLs of some common veterinary antibiotics in milk, such as β-Lactam, Sulfonamide, Tetracycline, Macrolide, and Aminoglycoside [24]. China established a risk assessment system for food safety China National Center for Food Safety in 2009, which can better support food safety monitoring and supervision [25]."

In conclusion, "on the positive side, in 2017, China developed a national veterinarian system for training, managing, and supervising animal health workers [54], this will help reduce antibiotic residues in milk. which will help to reduce antibiotic residues in milk."

Reference:

  1. Lu G, Chen Q, Li Y, Liu Y, Zhang Y, Huang Y, Zhu L. Status of antibiotic residues and detection techniques used in Chinese milk: A systematic review based on cross-sectional surveillance data. Food Res Int, 2021. 147: p. 110450.
  2. Wu, Y.-n., P. Liu, and J.-s. Chen. Food safety risk assessment in China: Past, present and future. Food Control, 2018. 90: p. 212-221.
  3. Zeng, Z., F. Yang, and L. Wang, Veterinary Drug Residues in China. Food Safety in China. 2017. p. 219-235.

Round 2

Reviewer 1 Report

Thank you for submitting this revised version of your manuscript.

We are in agreement for the definition of an udder and a quarter. However your manuscript (39 we can read inflammation of one or more udders… it is one or more quarters.

Author Response

Dear Reviewers,

Thank you for your letter again and for the new reviewers’ comments concerning our manuscript entitled “Association of mastitis and farm management to the contamination of antibiotics in bulk tank milk in Southwestern area, China” (ID: animals-1949262). The revised portions are marked in red in the paper. The main corrections in the paper and the responses to the reviewer’s comments are as flowing:

Response to reviewer 1

manuscript (line 39 we can read inflammation of one or more udders… it is one or more quarters.

Response: We have changed the first sentence to: “Bovine mastitis is defined as an inflammation of mammary gland of dairy cows”.

Reviewer 3 Report

Dear Authors,

made revisions have improved your manuscript considerably.

I recommend the publication of the manuscript in Animals.

Author Response

Dear Reviewers,

Thank you for your letter again and for the new reviewers’ comments concerning our manuscript entitled “Association of mastitis and farm management to the contamination of antibiotics in bulk tank milk in Southwestern area, China” (ID: animals-1949262). Thank you for you approved our manuscript.